# Zinc Chloride Enhances the Antioxidant Status, Improving the Functional and Structural Organic Disturbances in Streptozotocin-Induced Diabetes in Rats

**DOI:** 10.3390/medicina58111620

**Published:** 2022-11-10

**Authors:** Irina Claudia Anton, Liliana Mititelu-Tartau, Eliza Gratiela Popa, Mihaela Poroch, Vladimir Poroch, Ana-Maria Pelin, Liliana Lacramioara Pavel, Ilie Cristian Drochioi, Gina Eosefina Botnariu

**Affiliations:** 1Department of Pharmacology, Clinical Pharmacology and Algesiology, Faculty of Medicine, ‘Grigore T. Popa’ University of Medicine and Pharmacy, Universitatii St. 16, 700115 Iasi, Romania; 2Department of Pharmaceutical Technology, Faculty of Pharmacy, ‘Grigore T. Popa’ University of Medicine and Pharmacy, Universitatii St. 16, 700115 Iasi, Romania; 3Department of Family Medicine, Preventive Medicine and Interdisciplinarity, Faculty of Medicine, ‘Grigore T. Popa’ University of Medicine and Pharmacy, Universitatii St. 16, 700115 Iasi, Romania; 42nd Department of Internal Medicine, Faculty of Medicine, ‘Grigore T. Popa’ University of Medicine and Pharmacy, Universitatii St. 16, 700115 Iasi, Romania; 5Department of Pharmaceutical Sciences, Faculty of Medicine and Pharmacy, ‘Dunărea de Jos’ University, 800010 Galați, Romania; 6Department of Morphological and Functional Sciences, Faculty of Medicine and Pharmacy, ‘Dunărea de Jos’ University, 800010 Galați, Romania; 7Surgical Department, Faculty of Dental Medicine, University of Medicine and Pharmacy, 700115 Iasi, Romania; 8Department of Diabetes, Nutrition and Metabolic Disease, ‘Grigore T. Popa’ University of Medicine and Pharmacy, Universitatii St. 16, 700115 Iasi, Romania

**Keywords:** zinc chloride, streptozotocin, diabetes, rats, oxidative stress, functional and structural disturbances

## Abstract

*Background and Objectives*: Diabetes mellitus (DM) is a complex disease affecting the whole metabolic balance of the body and resulting in multiple organ complications: cardiovascular, neuronal, renal, etc. Our study focuses on investigating the effect of zinc chloride (Zn) on certain blood parameters suggestive for assessing the metabolic disturbances, the liver and kidney function, the oxidative stress and the immune defense capacity in experimental-induced DM with streptozotocin (STZ) and cholesterol in rats. *Materials and Methods*: The animals were assigned to three groups, as follows: Group 1 (Control): buffer citrate solution 0.1 mL/100 g body; Group 2 (STZ): 20 mg/kg body STZ and fat diet (10 g cholesterol/100 g diet); Group 3 (STZ+Zn): 20 mg/kg body STZ + 5 mg/kg body Zn chloride and the same fat diet. DM was induced by administering STZ in a single take daily, for three consecutive days, Zn and citrate buffer were administered orally for a month. The protocol was approved by the Ethics Committee of the University ‘Grigore T Popa’ Iasi, in agreement with the International Regulations about the handling of laboratory animals. *Results:* The use of STZ in rats fed with cholesterol was correlated with important weight gain, hyperglycemia, the intensification of the transaminases activity and the increase in serum alkaline phosphatase, cholesterol, triglyceride, urea, creatinine and in malondialdehyde. *Conclusions*: The treatment with Zn resulted in weight loss and a decrease in blood sugar in diabetic rats. Supplementation with Zn notably reduced oxidative stress, preserved the pancreatic architecture and restored the liver and kidney function and structure in STZ-induced DM in rats.

## 1. Introduction

The investigations conducted during this experimental research comply with the modern tendencies of scientific research that focus on establishing the effects of drugs or combinations of drugs in various pathological conditions which can be seen in medical practice, including diabetes mellitus (DM). 

DM is a heterogeneous syndrome characterized by a complex disruption in regulating the energetic metabolism of the body, which affects the use of carbohydrates, fats and proteins, as well as other metabolisms [1,2]. It is a chronic condition, inherited or acquired, caused by an imbalance between the body’s insulin demand and insulin availability, manifested by increased blood glucose levels [3,4]. 

The characteristic symptoms of hyperglycemia which lead to the diagnosis of diabetes, are: intense fatigue, loss of muscle mass due to decreased protein synthesis, dehydration, polydipsia, polyphagia, polyuria and glycosuria. In evolution, diabetes is associated with multiple pathological conditions, such as cardiovascular, neuropathy, renal and eye diseases [5,6]. The increased level of blood glucose induces an increase in the free radicals generation, thus resulting in high oxidative stress [7,8]. It is known that cellular oxidative processes are involved in the pathology of various diseases, including diabetes, being responsible for the development and progression of tissue alterations in different organs [9,10]. These injuries are the consequence of the imbalance between the oxidative and antioxidant mechanisms, as a result of the amplified activity of some pro-oxidant enzymes, as well as the excessive production of free radicals and/or decrease in the antioxidant defense [11,12].

Zinc (Zn) is the second metal in the body after iron and is the only metal found in all types of enzymes. Zn is present in all body tissues, mainly in the brain, muscles, bones, kidneys and liver. High concentrations are also found in the prostate and some parts of the eye [13]. The total amount of Zn in the body is 2–4 g. This trace element is necessary for normal growth and development in the prenatal period, in childhood and adolescence, in nerve excitability, synaptic plasticity and learning [14,15]. It also plays a role in the function of the immune system, in the regulation of apoptosis phenomena, in tissue repair, in protein synthesis and in cell division [16,17]. It participates in the synthesis of vitamin A in the liver, the metabolism of fats, carbohydrates and proteins in the body, oxidation processes (mediating superoxide dismutase functions), healing of wounds and burns, the mediation of inflammatory reactions and the synthesis of deoxyribonucleic acid [18,19].

The fundamental and varied roles of Zn in various processes carried out at the cellular level require that its release in tissues and at the cellular level, its distribution and intracellular availability are very strictly controlled and regulated. The involvement of Zn in mediating pathological processes in diabetes and especially metabolic disturbances, the associated organic structural and functional changes, is complex and not yet fully elucidated.

Zn also participates in the storage and processing of insulin in the human body. Literature data show that the administration of Zn supplements in patients with type 2 DM improves the symptoms, as a result of the decrease in cholesterol and glycated hemoglobin blood levels [20,21]. Normally, Zn is concentrated in the pancreatic beta cells, at the level of the dense nucleus of the insulin-secreting granules, being essential for the formation of insulin crystals and further insulin release, transport, storage and processing in the body [22,23,24].

The recommended dietary allowance (RDA) for Zn is 11 mg/day for men and 8 mg/day for women [25]. This is significantly below the LD50 value, which has been estimated to be 27 g/day in humans, based on a comparison with equivalent studies in rats and mice [26]. In general, the uptake of such an amount is unlikely, because approximately 225–400 mg of Zn has been determined to be an emetic dose. Immediate symptoms after the uptake of toxic amounts of Zn include abdominal pain, nausea and vomiting. Additional effects include lethargy, anemia and dizziness [27]. 

Clinical studies have revealed that in people with a copper deficiency, clinically objective, associated with anemia, neutropenia and the alteration of the immune defense capacity, the administration of small doses of Zn (100–300 mg Zn/day), but also of quantities much higher than the RDA (15 mg Zn/day), cause important disturbances of the LDL/HDL cholesterol ratio to occur. Even lower levels of Zn supplementation, closer in amount to the RDA, have been suggested to interfere with the utilization of Cu and iron and to adversely affect HDL-cholesterol concentrations [28].

High-dose Zn supplementation interferes with the uptake of copper (Cu). Hence, many of its toxic effects are in fact due to a copper deficiency. While systemic homeostasis and efficient regulatory mechanisms on the cellular level generally prevent the uptake of cytotoxic doses of exogenous Zn, endogenous Zn plays a significant role in cytotoxic events in single cells [27].

Copper is essential to life, being the third most abundant trace element in the body, after zinc and iron. However, in adults, an amount of 80,000 to 100,000 µg of copper intake from the outsource becomes toxic. Excess of Cu leads to the damage of the antioxidant enzyme function, oxidative modification of DNA and proteins, lipid oxidation, activates the redox-sensitive genes, suppresses the Zn consumption in the body and, also, induces anemia by interfering with iron transport. Cu is present in every tissue of the body and is stored mostly in the liver, but smaller amounts are found in the brain, heart, kidney and muscles. It can act as both an anti-oxidant and a pro-oxidant. Copper plays an important role in human metabolism, largely because it allows many fundamental and essential enzymes to function properly, and is also involved in the production of hemoglobin, myelin and melanin. Thus, Cu serves as a cofactor for enzymes, including the ferroxidase enzyme, cytochrome oxidase, superoxide dismutase, and amine oxidases. In gene regulation, it acts as a cofactor in a DNA-binding “copper fist” protein. [29]. Due to the involvement of Cu in multiple molecular processes, especially in the course of catalytic reactions, the disruption of the homeostasis of this element is responsible for the appearance of various disorders that cause disease. Although the role of Cu is not completely deciphered, it is accepted that the labile fractions of this ion manifest their toxic effects on the cell, following the interaction with DNA and the intensification of oxidative stress, as a result of a generation of free oxygen species [30]. 

It is known that in patients with diabetes, the content of Zn in the pancreas is very low and serum Zn levels are low, as a result of its loss due to excessive urination [31]. Low levels of Zn in blood plasma prevent the islets of Langerhans from secreting and producing insulin [32,33]. The influence of this bivalent cation on the cellular oxidative processes are not completely deciphered and somehow controversial. Various authors suggested that Zn has antioxidant activity in patients with type 2 DM, participating as an essential component of superoxide dismutase (SOD) and mediating glutathione metabolism [34,35,36,37]. Zn alleviates oxidative stress in diabetic patients by decreasing the lipid peroxidation rate [32], as well as by reducing chronic hyperglycemia, with the promotion of insulin receptor phosphorylation, as a result of increased glucose transport in cells [38]. Other authors indicate that the use of high doses of Zn in a diabetic patient with normal blood values of this microelement did not significantly influence the activity of antioxidant enzymes or the functionality of the circulatory system [39].

Although the current diabetes management is concentrated on decreasing blood sugar levels, therapy should aim at slowing down the evolution of the disease and at preventing a possible treatment failure [40,41]. Recent approaches of innovative treatment aim to achieve a pharmacological influence on the multiple physio-pathological disturbances present in type 2 DM. Given the heterogeneity of type 2 DM, the best management should include an early start on combined therapy using more drugs of various effects, combined with a change in the patient’s lifestyle [41].

Over time, various researchers have been concerned with investigating the effects of Zn supplementation, on the various pathological processes associated with DM. Multiple experimental research aimed at proving the favorable effects of some Zn salts on the control of blood sugar levels, on some biochemical parameters that reveal the functional capacity of some organs and on their structural alterations using different standardized animal models of type 1 [42,43] and 2 [32,44,45] DM. The influence of this trace element on the immune defense capacity and cellular oxidation processes in laboratory animals with diabetes was also targeted [46,47,48,49].

We aimed to assess the pharmacodynamic effects of Zn in diabetic rats. We intended to provide additional information on the effects of Zn chloride supplementation on the hematological, biochemical, immunological profile, oxidative stress and on the histopathological changes of some organs of streptozotocin-induced diabetes in rats.

## 2. Material and Method

### 2.1. Substances

Zn chloride (catalogue name 208086, molecular weight MW: 136.3), streptozotocin (STZ) (catalogue name S0130, ≥75% α-anomer basis, ≥98% HPLC, MW: 265.22), cholesterol (catalogue name C8667, ≥99%, MW: 386.65), buffer citrate (catalogue name C2488, citrate buffer solution 0.09 M) and distilled water (catalogue name 07-6061, MW: 18.02) were purchased from Sigma-Aldrich, Chemical Co. (Steinheim, Germany) (www.sigma-aldrich.com, accessed on 15 September 2022).

### 2.2. Animals

The research was performed on healthy, genetically unmodified white Wistar rats (weighing between 150 and 200 g) acquired through the biobase of ‘Grigore T. Popa’ University of Medicine and Pharmacy of Iaşi from the National Medical-Military Institute for Research and Development, Baneasa, Bucharest, Romania. 

The rats were brought into the experiment room one day before the experiment to adjust; they were kept in special cages (individually), under standard laboratory conditions (constant temperature 21 ± 2 °C relative humidity of 50–70% and an alternating illumination cycle (light/darkness = 12 h/12 h).

To prevent chronobiological influences, the investigations were always carried out between 8 am and 12 am. The animals were given tap water ad libitum, through special water fountains, and fed with standard granulated food bought from the National Medical-Military Institute for Research and Development, Baneasa, Bucharest, Romania. The daily intake of food was fully determined. The animals were weighed daily and their behavior was carefully observed (motility, ability to explore their environment, breathing, specific hygiene moves). On the day of the experiment, the animals did not receive any food or liquids.

### 2.3. The Experimental Protocol

The experimental study was conducted on rats which were induced with diabetes mellitus, taking the following steps: administration of a diet rich in cholesterol and experimentally induction of diabetes by administration of STZ; studying the effects of Zn chloride on the biochemical changes and oxidative stress in rats with diabetes and a cholesterol-rich diet.

#### 2.3.1. The Experimental Model of STZ-Induced Diabetes in Rats

For the experiment, the animals were randomly assigned to three groups of five rats each, as follows: Group 1 (coded Control): buffer citrate 0.1 mL/100 g body;Group 2 (coded STZ): 20 mg/kg body STZ and fat diet (10 g cholesterol/100 g diet), for 4 weeks;Group 3 (coded STZ+Zn): 20 mg/kg body STZ + 5 mg/kg body Zn chloride and fat diet (10 g cholesterol/100 g diet), for 4 weeks.

In order to reproduce type 2 DM as similarly as possible to the human condition, we adapted to our laboratory conditions an experimental model based on the use of a low multiple doses of STZ, in animals with a fatty diet, the dose which is sufficient to increase the plasma sugar level, without affecting insulin secretion [50]. Diabetes was induced by administering STZ in a single take daily, for three consecutive days; STZ, previously dissolved in buffer citrate (25 µg buffer 0.1 M of sodium citrate pH 4.5) was immediately administered intraperitoneally (20 mg/kg body weight). Zn chloride and citrate buffer were administered orally for a month, as a single daily dose. Diabetes was confirmed by high glucose values (over 250 mg/dL) in the blood of twelve-hour-fasted rats, measured using an Accu-Chek Go digital glucometer (Roche, Warsaw, Poland). The animals were weighed at the beginning of the experiment, after a week and a month, to see the dynamics of their weight. 

#### 2.3.2. Laboratory Investigations

Blood samples were drawn from the lateral tail vein, for lab tests three times during the experiment, respectively: before inducing diabetes (baseline), a week later, and four weeks later. For this purpose, the animal was positioned in a restrainer system with the tail immersed in a container with water warmed up at 42 °C for 40 s to dilate the lateral veins. One of the veins was marked and, after using the antiseptic chlorhexidine 2%, it was punctured with a fine catheter (the peripheral vein with a butterfly catheter), slanted along the blood vessel, from approximately 5 cm from the tip of the tail. A volume of 0.3 mL of blood was collected from each animal with a syringe attached to the catheter [51]. 

The blood samples were analyzed for the following hematological and biochemical parameters: number of red blood cells (RBC); leukocyte formula (the percentage values of polymorphonuclear neutrophils—PMN, lymphocytes—Ly, monocytes—M, eosinophils—E, basophils—B); serum sugar level; activity of alanine-aminotransferase (ALT), aspartate-aminotransferase (AST), bilirubin, alkaline phosphatase; blood level of cholesterol and its fractions (low-density lipoprotein—LDL-cholesterol, high-density lipoprotein—HDL-cholesterol), triglycerides; urea and creatinine values; complement activity and peripheral blood PMN phagocytosis ability (Nitro blue tetrazolium test—NBT test); activity of malondialdehyde (MDA) and SOD.

The complete blood cell count was conducted on blood samples with anticoagulant (EDTA—tripotassium/dipotassium/disodium), collected in purple/pink vacutainers—K3 EDTA, using the hematology, lab animal automatic analyzer, HemaVet 950FS for veterinary use (Drew Scientific, Inc, Boston, MA, USA). For the biochemical data assessment, the VITROS 750 XRC Analyser was used (Alphasoft, Bochum, Germany), with specific reactors from Johnson & Johnson (Johnson, New Haven, CT, USA).

The number of erythrocytes was estimated with the help of the automatic hematology analyzer during their passage through an orifice through which they are directed in a single row by the hydrodynamic focusing method. Differential white blood cell counts were evaluated in all rats through the direct current detection technique, and percentages were assessed using the specific analyzer [52].

The transaminases activity was measured on blood collected in vacutainers without anticoagulants by the IFCC (International Federation for Clinical Chemistry) standardized kinetic method with pyridoxal phosphate. The serum levels of biochemical parameters were measured on blood samples collected without anticoagulant, using spectrophotometric methods (colorimetric for liver transaminases, total bilirubin, alkaline phosphatase and creatinine; colorimetric/enzymatic for triglycerides, total cholesterol and its fractions; respectively, kinetic for urea) [53].

The effects of the substances tested on the immune system were assessed by measuring the serum complement level and peripheral blood PMN phagocytose ability. The level of serum complement was determined using the Hartmann–Brećy method, which is based on the fact that a serum complement causes hemolysis in sensitive erythrocytes after connecting specific antibodies, measuring its activity after producing 50% hemolysis [54]. 

The NBT test shows changes in metabolism taking place during phagocytosis, quantitatively correlated with the intensity of this process, which is based on reducing nitroblue tetrazolium (NBT). After mixing the whole of the blood with NBT in a buffer solution of phosphate and physiologic serum, Giemsa coloration smears were made, and the microscopic exam helped to establish the percentage of NBT-positive neutrophils [55].

The effect on oxidative stress was assessed by measuring the activity of specific enzymes: MDA and SOD. The plasma level of MDA was quantified by high-performance liquid chromatography (HPLC) with fluorescence detection, based on the method of thiobarbituric acid [56] using a specific kit (catalogue code ABIN772058) purchased from Redox, Bucharest, Romania. The SOD activity was spectrophotometrically assessed using special kits from the RANSOD kit (catalogue code19160-1KT-F) of RANDOX Laboratories Ltd. (Crumlin, UK), by the colorimetric method with xanthine and xanthine oxidase. Xanthine-xanthine oxidase is used to generate O2•−, and the reduction of nitroblue tetrazolium is used as an indicator of the production of O2•−. Xanthine and xanthine-oxidase were used to generate superoxide anionic radicals which react to chloride of 2-(4-iodophenyl)-3-(4-nitrophenol)-5-phenyltetrazolium, this being a quantitative method. SOD inhibits reactions by transforming superoxide radicals into oxygen [57].

#### 2.3.3. Histologic Examination

After sacrificing the animals, fragments of organs were harvested (liver, pancreas, kidneys, cord) for observing the histopathological changes. After collecting them, the fragments were fixed in 10% formalin, then included in paraffin (Automatic Processor Device Leica TP 1020, Nussloch, Germany) and subsequently partitioned at 5 μm distance with a microtome (Slee-5062 Semi-automatic Apparatus, Nieder-Olm, Germany) to obtain thin tissue cross-slices. The sections were stained using Masson’s trichrome method, and examined with a photon microscope (Leica DM 1000, Wetzlar, Germany) equipped with a digital image acquisition system. A digital histological camera (Leica 5 mpx, full HD, Nussloch, Germany) and LAS software, version 2016 (Nussloch, Germany), were used to capture the images.

### 2.4. Ethical Aspects of the Research

The animal studies were conducted after approval from the University’s Ethics Committee (Ethics Notice No. 30/14.01.2021; Authorization from the Sanitary Veterinary Direction no. 30/24.02.2021), in full agreement with the International Ethics Regulations applied to working with lab animals.

The research methodology was carried out in agreement with international and national standards [58].

The experiments were conducted scientifically, observing pre-established protocols, using scientifically approved tests, following the 3R rule: replacement, reduction (reduce the suffering of the animal) and refinement (refine the method to reduce the pain of the animals).

### 2.5. Statistical Analysis of Data

The data were expressed as mean ± standard deviation (S.D.) of the average values, for each substance examined, and were statistically processed using the SPSS software version 17.0 for Windows and EXCEL application (IBM, New York, NY, USA). The statistical analysis allows us to understand the significance of the differences between the tested substances and the control group. Additionally, we used post hoc Tukey and Newman-Keuls tests for multiple comparisons, to divide the animal lots into significance groups, depending on the pharmacodynamic effect level and to rank the intensity of the action of the substances studied. In all situations, the values of the *p* coefficient below 0.05, respectively, 0.01, were considered statistically significant compared to the control group.

## 3. Results

No essential differences in body weight were noticed between the control group, STZ and STZ+Zn before the administration of the substances (Figure 1).

Inducing diabetes with STZ caused a progressive increase in the rats’ bodyweight over time, which was statistically relevant compared to the lot which received buffer citrate, after only one month (* *p* < 0.05) during the experiment. At the time of examination, the animals’ weight was considerably (**♦**
*p* < 0.05) higher than at the start of our experiment (Figure 1).

Using Zn on rats given STZ and cholesterol lowered the increase in body weight; the figures after weighing were not significantly different from the buffer citrate lot, or compared to those at the start of the experiment, at both times of observation (Figure 1).

No obvious differences in the number of RBC, in animals from the STZ and STZ+Zn groups, compared to the animals treated with buffer citrate were shown during the experiment (Table 1).

The use of STZ and STZ+Zn did not result in statistically significant variations in the percentage values of the leukocyte formula components (PMN, Ly, E, M, B) compared to the control lot at the three moments of the examination (Table 2).

At the beginning of the experiment, no changes in blood sugar levels between the three groups of animals were evidenced (Figure 2). The intraperitoneal STZ injection was associated with a continued rise in blood glucose values over time, which is statistically significant compared to the buffer citrate lot, after a week (** *p* < 0.01), as well as after a month (** *p* < 0.01) in the experiment. At these times the blood sugar levels were substantially higher (**♦♦**
*p* < 0.01) compared to the beginning of the experiment (Figure 2). Zn supplementation in STZ rats slowed down the increase in blood sugar levels, which were still slightly higher, but not significantly compared to the control lot, after one week. After a month, however, we noticed that the serum value of blood sugar decreased, becoming insignificant compared to the control group (Figure 2).

No substantial differences in ALT and AST serum levels in animals from the control, STZ and STZ+Zn groups were noted before the administration of the tested substances (Table 3). STZ administration resulted in increased ALT and AST activity, which proved to be significant (* *p* < 0.05) compared to the buffer citrate lot after only four weeks. At the same time of evaluation, ALT and AST serum levels were considerably (**♦**
*p* < 0.05) higher compared to the levels at beginning of the experiment (Table 3). Using Zn reduced blood AST and ALT values, which reached levels comparable to the control lot after a week, as well as after four weeks in the experiment (Table 3).

Before the administration of the test substances, there were no significant differences in the serum levels of total bilirubin and alkaline phosphatase in the three studied substances (Table 4). Using STZ and STZ+Zn, respectively, in rats fed with a high-fat diet induced a minor decrease in bilirubin, which was insignificant compared to the control group at the two moments of the experiment (Table 4). The treatment with STZ+Zn, but not with STZ was accompanied by a considerable (* *p* < 0.05) increase in alkaline phosphatase activity compared to the control group, but only after one month in the experiment (Table 4). At this moment of evaluation, its blood value was statistically significant (**♦**
*p* < 0.05) compared to the baseline.

No essential changes in the values of total cholesterol, of LDL-cholesterol, HDL-cholesterol and triglycerides was observed in the studied groups, at the beginning of the experiment (Table 5).

In animals with the fat diet and STZ, we noticed a marked increase in the level of total cholesterol, compared to the buffer citrate group, after one week (* *p* < 0.05), as well as after four weeks (* *p* < 0.05). A significant (* *p* < 0.05) increase in the blood values of triglycerides, compared to the control group, was detected only after one month. At this moment of observation, the levels of triglycerides were considerably increased (**♦**
*p* < 0.05), compared to time zero (Table 5). Furthermore, a decrease in the serum HDL-cholesterol was shown, which became obvious (* *p* < 0.05) compared to the control after four weeks. The values of this cholesterol fraction were substantially (**♦**
*p* < 0.05) lower compared to the start of the experiment, at the same time of evaluation (Table 5). Zn supplementation in rats with STZ and a fat diet was associated with a moderate increase in cholesterol, LDL-cholesterol and triglycerides, and, respectively, with a slight decrease in HDL-cholesterol, but without statistical significance compared to the control group, as well as the initial moment in the experiment (Table 5).

Before the administration of the substances, no relevant differences in urea and creatinine serum levels, between the tested groups and control animals were shown (Table 6). Although, throughout the biochemical monitoring period, a gradual alteration in the renal function parameters was noted; their values became significant compared to the control group (* *p* < 0.05), respectively, to baseline (**♦**
*p* < 0.05) after four weeks in the experiment (Table 6). 

The assessment of the serum complement and the NBT test values did not reveal major differences between the tested groups at time zero (Table 7). After the administration of STZ and STZ+Zn, respectively, in animals with the fat diet, we noticed a modest increase in complement activity, and a slight reduction in the phagocytose capacity of peripheral blood PMN after one and then four weeks, but without statistical relevance compared to the buffer citrate group and also to the baseline (Table 7).

No important changes in serum SOD and MDA values were observed in these three studied groups at the beginning of the experiment (Table 8). The treatment with STZ produced a gradual decrease in SOD activity, respectively, and an intensification in MDA activity, statistically relevant compared to the control (* *p* < 0.05), as well as to the baseline (**♦**
*p* < 0.05) after one month (Table 8). The use of the combination STZ+Zn reduces the modification in levels of these two enzymes, their blood values being comparable with those of the control group and with the baseline, at both moments of investigation (Table 8).

The histopathological investigation revealed that in the control animals the pancreas had a regular architecture, with insular cells interspersed between whole acinar cells, which are grouped in lobes. Pancreatic lobes are separated by an intact septum of intra- and interlobular connective tissue (Figure 3). In rats treated with STZ, we noticed an important alteration of the exocrine, as well as endocrine components. The beta-pancreatic cells were very rare, the acinar cells appeared expanded in volume (arrows), and contained small vacuoles(v). The interlobular channels were coated with flat epithelium (Figure 3). The histopathological exam of the pancreatic fragments collected from the STZ+Zn group revealed that the atrophy of acinar cells was slighter than in the STZ group, with a clearer line between exocrine and endocrine areas of the pancreatic tissue being detected. A tendency to regenerate the insular areas was observed, and the vacuoles at the base of the acinar cells were few and small (Figure 3).

The hepatic structure appeared normal in animals from the control group, in the central lobular vein as well as nearby hepatocytes, whose architecture is not significantly changed (Figure 4). The microscopic examination of the samples obtained from the hepatic tissue of STZ rats indicates liver steatosis objectified by the presence of numerous small lipid vacuoles (v) inside the hepatocytes. The hepatic cells showed moderate disturbances in their structure, their cytoplasm being slightly darker and the voluminous nuclei have prominent nucleoli. The presence of a minor periportal fibrosis(arrows) and sinusoidal distension were also observed (Figure 4). In the STZ+Zn group, we noticed a lower degree of microvacuolar hepatic steatosis with slight changes in hepatic cell conformation (Figure 4).

The microscopic evaluation of the kidney fragments from rats treated with buffer citrate showed a normal structure of both glomerular and tubular areas (Figure 5). STZ administration caused important changes in kidney architecture, characterized by a medium degeneration of cortical area and tumefied nephrocytes in the tubular epithelium (T), with dark cytoplasm and hyperchromatic nuclei. Intense glomerular congestion and infiltration of mononuclear cells were also present (Figure 5). The STZ+Zn group showed moderate congestion of interstitial capillaries and a slight infiltration of mononuclear cells (arrow) (Figure 5).

A histopathological examination of the slides made from myocardia fragments revealed a normal structure of the smooth muscle fiber and blood vessels in the control animals (Figure 6). In STZ rats, moderate to severe myocardial congestion, consisting of moderate edema, polymorphonuclear infiltration and blood overcharge of interstitial capillaries were detected (arrows). However, there were no visible morphological changes in myocardiocyte structure (Figure 6). The STZ+Zn group only showed slight myocardial congestion, without other significant architectural changes (Figure 6).

## 4. Discussion

Increased plasma concentrations of Zn have been associated with reduced cardiovascular risk in people with diabetes [59,60], but its use in the prevention of cardiovascular disease remains unclear. The administration of the essential trace mineral (Zn) has been proposed as a strategy against type 2 DM; however, there are currently no official recommendations based on solid systematic reviews to support the protective effects of Zn against cardio-metabolic conditions.

Literature data revealed that the favorable effects of Zn for reducing cardiovascular risk are much more pronounced among patients with additional risk factors such as type 2 DM [20]. There is strong evidence that improving plasma Zn levels in patients with diabetes could be an effective strategy for reducing the complications of diabetes, including cardiovascular disease. The researchers claim that an optimal Zn level in the body through supplementation or diet was associated with the improvement of biological markers that assess the progression of type 2 DM (glycemia, plasma lipid profile) [61,62]. The reduction in secondary complications of this condition was also noted [34]. These data are not enough to issue new universal recommendations for the prevention of cardiovascular diseases and complications of type 2 diabetes. Additional studies are needed to establish the mechanism by which Zn induces these protective effects, the safe doses at which these effects appear, the bioavailability and ideal sources of Zn and information that cannot be obtained based on observational studies alone.

Many studies conducted on laboratory animals have focused on highlighting a correlation between Zn levels in the body and diabetes conditions in different stages. These researches use animal experimental models based on induced diabetes in rats (with STZ), with a Zn supplementation of 6 to 8 weeks duration. STZ is one of the diabetogenic agents most used to induce this metabolic disturbance in rodents, because it causes the alteration of pancreatic beta cells, with a reduction in insulin levels and a decrease in glucose tolerance. The mechanisms by which STZ produces diabetes involve the deterioration of glucose oxidation and the disruption of the function of beta cells, which no longer respond to glucose. STZ shows a high affinity for lipids, being absorbed very quickly through the plasmatic membrane, and accumulated in the beta Langerhans pancreatic cells, as well as in the liver and kidney tissue, with cytotoxic effects. At the cellular level, STZ produces the glycosylation of proteins and DNA alteration, which consequently reduces cellular energy stores. The intensity of these effects correlates positively with the dose, the small repeated doses, producing limited structural and functional and metabolic disturbances [63].

Barman et al. demonstrated good effects on hyperglycemia and hypoinsulinemia in diabetic rats who had received Zn supplementation in high doses for 6 weeks. These positive effects were objectified by observing certain parameters that undergo critical changes in diabetes: plasma albumin levels and urea and creatinine levels, which were significantly lower in rats receiving Zn compared to the control. However, these effects were obtained by using high doses of Zn supplementation (up to 10 times the normal level) [48]. By supplementation of high doses of Zn (orally, 100 mg/kg body weight Zn sulphate), other research could demonstrate the significant protection of stomach mucosa of diabetic rats, by microscopic examination and testing of enzyme levels involved in stomach tissue metabolism [46].

Literature data show that low Zn levels contribute to the induction of oxidative stress; thus, the involvement of Zn in oxidative processes in the human body has been intensively studied and discussed over the years. The physiopathological processes through which this bivalent cation mediates the protection of cells against the negative action of oxygen free radicals involve its effects of ensuring the preservation of sulfhydryl groups in proteins [64]. Moreover, by stabilizing lipids and proteins, Zn can protect the cell membrane and protein macromolecules from oxidative damage [65]. As a component element of the antioxidant enzyme SOD, Zn has an essential involvement in improving oxidative status [36] and in reducing the rate of lipid peroxidation [38]. Zn decreases the generation of reactive oxygen species (ROS) through the following mechanisms: inhibits nicotinamide adenine dinucleotide phosphate oxidase and, together with copper, it is part of SOD induces metallothionein, which reduces hydroxyl radicals. The high degree of protection of Zn on the stability of the cell membrane constitutes a solid barrier against free radicals [66].

In diabetes mellitus, oxidative stress aggravates the damaging effects on the organs, such as the heart, liver, kidneys, stomach, etc. The antioxidant effect, however, has also been demonstrated at lower doses (e.g., 5 to 10 mg/kg body weight), to show that Zn significantly decreased levels of MDA, but also of fasting blood glucose and triglycerides and in diabetic rats [47]. 

Higher doses of Zn supplementation have been demonstrated to improve cardiovascular tissue architecture in diabetic rats, with cardio-protective effects shown by a significant alteration of mRNA expressions of fibrosis and factors inducing apoptosis. By Zn supplementation, some markers indicating cardiac oxidative stress, such as the 4-hydroxy-2-nonenal could be improved by a high percentage (68%) [48]. The protective effects of Zn sulphate in cardiomyopathy induced by diabetes in rats were also quantified by the oxidative stress marker glucose-regulated protein-78, showing positive effects on the cardiac tissue by inhibition of autophagy and by decreasing endoplasmic reticulum stress [67].

The role of Zn supplementation was also studied, in the alterations of kidney tissue of rats with experimentally induced diabetes. It was shown that dietary Zn protected the renal cortical architecture alteration in rats with diabetes induced with STZ and a fat diet [68]. In our study, low doses of STZ and a cholesterol diet were used to obtain a DM model with metabolic disorders similar to human type 2 DM. In addition, STZ is easy to purchase and is not expensive, and DM type 2 appears after a relatively short time.

In our experimental conditions, the administration of 20 mg/kg body STZ, for three consecutive days, and cholesterol for one month, in white Wistar rats, induced DM, objectified by the significant increase in blood glucose values, but also of the animals’ weight.

The present research revealed no substantial changes in the number of erythrocytes or in the percentage of neutrophils, lymphocytes, eosinophils, monocytes and basophils in diabetic rats and the STZ+Zn group when compared to control animals. Moreover, the percentage of these leukocyte formula components was not significantly modified at the two time moments of evaluation, compared to the baseline. Literature data reported considerable variations in the percentages of these elements in type 1 diabetes, induced by a single high dose of STZ in rats [69].

Concerning the effects on hematological parameters of low doses of STZ, the communicated results were only few and controversial. Some authors demonstrated a significant decrease in both neutrophil and lymphocyte percentages, as well as in the number of RBC in rats subjected to 15 mg/kbw STZ compared to normal rats [70]. Other researchers proved a marked increase in the neutrophils percentage, and a diminution in lymphocytes percentage and in the RBC number in type 2 diabetes experimentally induced with 35 mg/kg STZ and fat diets in rats when compared with animals treated with buffer citrate [71].

It was pointed out that, in diabetic animals, the values of urea and creatinine increased considerably after four weeks, a fact that demonstrates the disturbance of renal function. These findings were correlated with the presence of characteristic lesions of renal tissue structural alteration by histopathological examination. Laboratory investigations revealed an intensification of the activity of liver enzymes and alkaline phosphatase after one month. In addition, the important disturbance of lipid metabolism, a significant increase in the serum levels of total cholesterol, HDL-cholesterol and triglycerides was noted. Laboratory analysis did not evidence the modifications in the serum levels of LDL-cholesterol, even a slight increase was noted after 4 weeks in diabetic rats, compared to the control group. Total bilirubin values in blood were not extensively influenced by the low doses of STZ, which probably induced only moderate alterations in liver functions. The identification of changes in liver conformation was added to these biochemical disturbances, suggesting a moderate impairment of the liver function and architecture. The characteristic elements of liver injury can be explained by the fact that this tissue is insulin-dependent, being decisively involved in both lipid and carbohydrate metabolism [72]. The STZ-induced type 2 DM in rats with a high-fat diet was not accompanied by essential changes in the activity of serum complement and in the phagocytic capacity of peripheral neutrophils, which denotes the fact that, with the dose used, this diabetogenic agent does not profoundly alter the immune defense capacity. The experimental induction of DZ with STZ was associated with an escalation of oxidative stress; after one month, a substantial increase in blood values of MDA values and also, a decrease in blood SOS levels were evidenced. The treatment with Zn chloride, during the four weeks in which the rats were exposed to STZ and a fat diet, had the result of the amendment of biochemical parameters disturbances in evaluating liver and kidney function, and of those involved in cellular oxidation processes. These effects are positively correlated with the improvement of pancreatic, hepatic, renal and myocardial lesions in diabetic rats. The research carried out clearly shows the fact that the treatment with cholesterol and 15 mg/kg STZ, 3 consecutive days, caused DM type 2 in rats, characterized by weight gain and hyperglycemia, and was correlated with moderate structural and functional liver and renal disturbances, as well as an intensification of oxidative stress, changes that could be present in mild and moderate phases of the disease.

The fact that Zn supplementation contributes to reducing weight, normalizing blood glucose levels, reducing oxidative stress and ameliorating liver and kidney damage constitutes an argument regarding the protective role of this trace element in mild and moderate forms of DM. Regarding the protective influence of Zn in severe forms of DM, further in-depth studies are required.

## 5. Conclusions

Three sub-toxic daily doses of STZ and high-fat feeding were used to induce type 2 DM in rats, without a major influence on insulin secretion and massive pancreatic destruction. 

In our study, we proved that the treatment with Zn chloride resulted in weight loss and a decrease in blood sugar in diabetic rats, which suggests Zn protection in this illustrative animal model of human type 2 DM. The supplementation with Zn notably reduced oxidative stress, preserved the pancreatic architecture, and restored the liver and kidney function and structure in STZ-induced DM in rats.

We can conclude that Zn chloride improved the antioxidant defense capacity, as well as the functional and structural organic deteriorations in diabetic rats, which makes us appreciate that its potential use as a supplement for the treatment of DM is worthy of further in-depth research. 

The results obtained by us are of great importance in medical practice, regarding the favorable effects of Zn in slowing down the evolution of the disease and the occurrence of organic complications in diabetes.

## Figures and Tables

**Figure 1 medicina-58-01620-f001:**
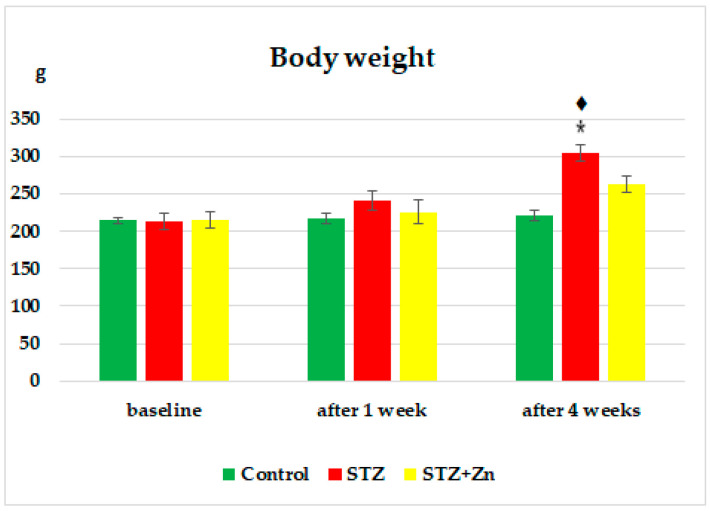
Effects of Zn on the body weight of diabetic rats. The values are presented as mean ± S.D. for 5 animals in a group (STZ—streptozotocin, STZ+Zn—steptozotocin + zinc). * *p* < 0.05 versus control; **♦**
*p* < 0.05 versus baseline.

**Figure 2 medicina-58-01620-f002:**
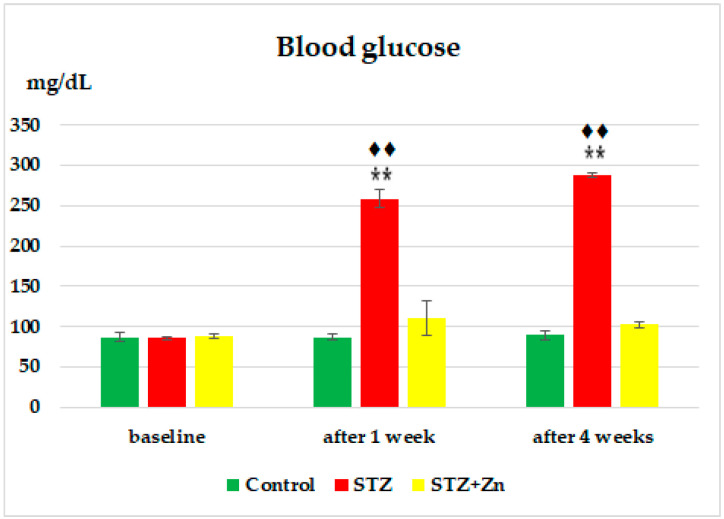
The effects of Zn on the glycemia values in diabetic rats. The values are presented as arithmetic average ± S.D. for five animals in a group. ** *p* < 0.01 versus control; **♦♦**
*p* < 0.01, versus baseline.

**Figure 3 medicina-58-01620-f003:**
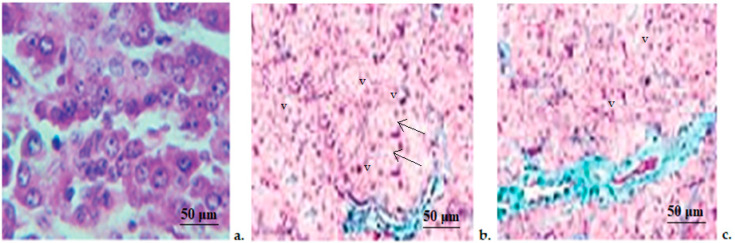
Pancreatic histopathological exam. (**a**) Control group; (**b**) STZ group; (**c**) STZ+Zn group. (arrow—acinar cells, v—vacuoles). Masson trichrome stain × 20.

**Figure 4 medicina-58-01620-f004:**
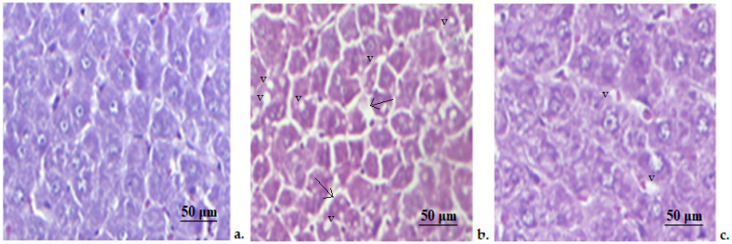
Liver histopathologic exam. (**a**) Control group; (**b**) STZ group; (**c**) STZ+Zn group. (v—vacuoles). Masson trichrome stain × 20.

**Figure 5 medicina-58-01620-f005:**

Kidney histopathologic exam. (**a**) Control group; (**b**) STZ group; (**c**) STZ+Zn group. (arrow—infiltration of mononuclear cells, T—tubular epithelium). Masson trichrome stain × 20.

**Figure 6 medicina-58-01620-f006:**
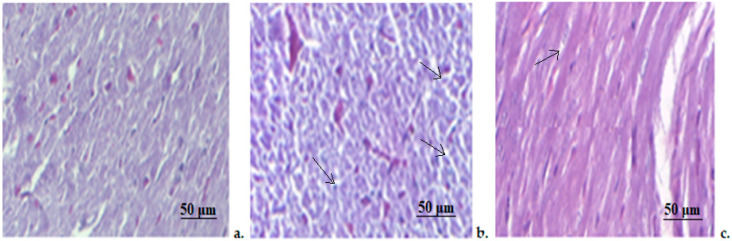
Myocardial histopathological exam. (**a**) Control group; (**b**) STZ group; (**c**) STZ+Zn group. (arrow—polymorphonuclear infiltration). Masson trichrome stain × 20.

**Table 1 medicina-58-01620-t001:** The effect of Zn on number of RBC in diabetic rats. The values are presented as mean ± S.D. for 5 animals in a group.

	Moment of Evaluation	RBC (mil/μL)
Control	Baseline	8.56 ± 1.27
	after 1 week	8.43 ± 1.65
	after 4 weeks	8.38 ± 1.22
STZ	Baseline	8.64 ± 1.32
	after 1 week	8.48 ± 1.47
	after 4 weeks	8.35 ± 1.18
STZ+Zn	Baseline	8.61 ± 2.15
	after 1 week	8.52 ± 1.25
	after 4 weeks	8.46 ± 1.30

**Table 2 medicina-58-01620-t002:** The effect of Zn on percentages of the leukocyte formula elements in diabetic rats. (PMN—polymorphonuclear neutrophils, Ly—lymphocytes, M- monocytes, E—eosinophils, B—basophils). The values are presented as mean ± S.D. for five animals in a group.

	Moment of Evaluation	Leukocyte Formula Elements (%)				
		PMN	Ly	E	M	B
Control	Baseline	27.4 ± 4.25	68.4 ± 7.39	0.3 ± 0.01	3.7 ± 0.03	0.2 ± 0.01
	after 1 week	27.1 ± 4.11	68.6 ± 8.42	0.5 ± 0.03	3.6 ± 0.05	0.2 ± 0.01
	after 4 weeks	27.2 ± 4.67	68.9 ± 8.33	0.4 ± 0.03	3.3 ± 0.05	0.2 ± 0.03
STZ	Baseline	26.8 ± 3.83	69.2 ± 8.27	0.4 ± 0.03	3.4 ± 0.01	0.2 ± 0.01
	after 1 week	27.5 ± 5.17	68.5 ± 7.53	0.3 ± 0.01	3.5 ± 0.03	0.2 ± 0.01
	after 4 weeks	27.3 ± 4.11	68.7 ± 8.42	0.3 ± 0.03	3.5 ± 0.05	0.2 ± 0.01
STZ+Zn	Baseline	26.7 ± 3.45	69.3 ± 8.67	0.3 ± 0.03	3.5 ± 0.05	0.2 ± 0.01
	after 1 week	26.9 ± 4.30	68.8 ± 9.22	0.4 ± 0.01	3.7 ± 0.05	0.2 ± 0.01
	after 4 weeks	27.0 ± 4.55	68.9 ± 8.17	0.5 ± 0.01	3.4 ± 0.03	0.2 ± 0.03

**Table 3 medicina-58-01620-t003:** The influence of Zn on ALT and AST activity in diabetic rats. The numbers are presented as mean value ± S.D. for five animals in a group. (ALT—alanine-aminotransferase, AST—aspartate-aminotransferase) * *p* < 0.05 versus control; **♦**
*p* < 0.05 versus baseline.

	Moment of Evaluation	ALT (U/L)	AST (U/L)
Control	Baseline	40.54 ± 4.29	91.64 ± 7.27
	after 1 week	42.72 ± 5.55	90.87 ± 7.43
	after 4 weeks	41.36 ± 4.17	91.25 ± 6.67
STZ	Baseline	40.63 ± 4.83	90.45 ± 6.33
	after 1 week	48.24 ± 4.45	113.39 ± 7.45
	after 4 weeks	69.33 ± 5.22 *♦	124.63 ± 7.18 *♦
STZ+Zn	Baseline	41.48 ± 4.67	91.55 ± 6.83
	after 1 week	48.51 ± 5.13	100.36 ± 6.33
	after 4 weeks	45.75 ± 5.05	99.48 ± 6.29

**Table 4 medicina-58-01620-t004:** The influence of Zn on blood levels of bilirubin and alkaline phosphatase in rats with experimentally induced diabetes. The numbers are presented as average ± S.D. for five animals in a group. * *p* < 0.05 versus control; **♦**
*p* < 0.05 versus baseline.

	Moment of Evaluation	Total Bilirubin (mg/dL)	Alkaline Phosphatase (U/L)
Control	Baseline	0.82 ± 0.03	121.56 ± 7.22
	after 1 week	0.79 ± 0.01	123.25 ± 8.13
	after 4 weeks	0.80 ± 0.03	122.32 ± 7.67
STZ	Baseline	0.85 ± 0.07	120.63 ± 7.43
	after 1 week	0.77 ± 0.05	127.45 ± 6.83
	after 4 weeks	0.76 ± 0.03	138.52 ± 8.33 *♦
STZ+Zn	Baseline	0.81 ± 0.01	121.39 ± 7.67
	after 1 week	0.78 ± 0.05	124.67 ± 7.55
	after 4 weeks	0.79 ± 0.03	123.84 ± 7.27

**Table 5 medicina-58-01620-t005:** The influence of Zn on the serum levels of total cholesterol, LDL-cholesterol, HDL-cholesterol and triglycerides in diabetic rats (LDL-cholesterol—low-density lipoprotein cholesterol, HDL-cholesterol—high-density lipoprotein –cholesterol). The values are presented as mean value ± S.D. for five animals per lot. * *p* < 0.05 versus control; **♦**
*p* < 0.05 versus baseline.

	Moment of Evaluation	Total Cholesterol (mg/dL)	LDL-Cholesterol (mg/dL)	HDL-Cholesterol (mg/dL)	Triglycerides (mg/dL)
Control	Baseline	65.48 ± 5.67	23.53 ± 1.11	45.44 ± 2.43	51.82 ± 4.25
	after 1 week	64.33 ± 5.21	21.61 ± 1.43	46.62 ± 2.17	52.25 ± 4.67
	after 4 weeks	67.26 ± 6.30	24.85 ± 1.05	45.73 ± 1.55	52.16 ± 5.43
STZ	Baseline	64.82 ± 4.13	22.48 ± 1.17	45.35 ± 2.21	52.44 ± 5.21
	after 1 week	78.63 ± 6.45 *♦	25.32 ± 1.55	35.36 ± 2.67	60.59 ± 5.33
	after 4 weeks	80.54 ± 6.33 *♦	29.64 ± 2.13	34.28 ± 2.05 *♦	63.78 ± 6.17 *♦
STZ+Zn	Baseline	66.42 ± 4.55	21.71 ± 1.22	46.14 ± 2.33	51.34 ± 4.83
	after 1 week	74.37 ± 5.05	26.12 ± 1.35	36.65 ± 1.29	56.26 ± 4.79
	after 4 weeks	75.76 ± 5.83	27.24 ± 2.33	37.30 ± 1.43	54.26 ± 5.55

**Table 6 medicina-58-01620-t006:** The effects of Zn on serum levels of urea and creatinine in diabetic rats. The numbers are presented as arithmetic average ± S.D. for five animals in a group. * *p* < 0.05 versus control; **♦**
*p* < 0.05 versus baseline.

	Moment of Evaluation	Urea (mg/dL)	Creatinine (mg/dL)
control	Baseline	40.12 ± 3.33	0.13 ± 0.01
	after 1 week	42.05 ± 3.41	0.14 ± 0.01
	after 4 weeks	42.28 ± 13.17	0.12 ± 0.03
STZ	Baseline	40.26 ± 3.17	0.13 ± 0.03
	after 1 week	47.55 ± 4.43	0.15 ± 0.01
	after 4 weeks	59.83 ± 6.13 *♦	0.26 ± 0.03 *♦
STZ+Zn	Baseline	39.45 ± 3.67	0.12 ± 0.01
	after 1 week	43.24 ± 3.55	0.14 ± 0.01
	after 4 weeks	44.58 ± 3.82	0.14 ± 0.03

**Table 7 medicina-58-01620-t007:** The effect of Zn on the activity of serum complement activity and the NBT test values in diabetic rats. (NBT—nitroblue tetrazolium, UCH50—50% haemolytic complement activity). The values are presented as mean value ± S.D. for five animals in a group.

	Moment of Evaluation	Complement (UCH_50_)	NBT (%)
Control	Baseline	53.50 ± 4.43	13.33 ± 1.55
	after 1 week	54.42 ± 4.17	14.17 ± 1.83
	after 4 weeks	54.38 ± 4.30	14.56 ± 2.67
STZ	Baseline	53.83 ± 4.29	13.52 ± 1.43
	after 1 week	62.94 ± 5.13	11.21 ± 1.17
	after 4 weeks	63.78 ± 5.22	11.05 ± 1.29
STZ+Zn	Baseline	53.25 ± 3.41	14.78 ± 1.22
	after 1 week	55.46 ± 4.88	13.24 ± 1.05
	after 4 weeks	56.74 ± 4.55	12.53 ± 1.13

**Table 8 medicina-58-01620-t008:** The influence of Zn on the activity of SOD and MDA in diabetic rats. The values are presented as mean value ± S.D. for five animals per lot. (SOD—superoxide dismutase, MDA—malondialdehyde). * *p* < 0.05 versus control; **♦**
*p* < 0.05 versus baseline.

	Moment of Evaluation	SOD (U/mg Protein)	MDA (nMol/mg Protein)
Control	Baseline	104.48 ± 6.43	22.54 ± 1.35
	after 1 week	105.63 ± 6.55	23.27 ± 2.11
	after 4 weeks	106.39 ± 7.27	22.65 ± 1.67
STZ	Baseline	105.22 ± 6.43	22.83 ± 1.43
	after 1 week	98.33 ± 4.51	29.48 ± 2.13
	after 4 weeks	79.85 ± 5.13 *♦	32.67 ± 1.25 *♦
STZ+Zn	Baseline	105.34 ± 5.83	21.25 ± 1.33
	after 1 week	103.62 ± 6.67	26.71 ± 2.21
	after 4 weeks	104.29 ± 6.21	25.44 ± 1.17

## Data Availability

Not applicable.

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
