# Peer review of "Zinc Chloride Enhances the Antioxidant Status, Improving the Functional and Structural Organic Disturbances in Streptozotocin-Induced Diabetes in Rats"

_medicina, 2022, doi:10.3390/medicina58111620_

Round 1

Reviewer 1 Report

The authors have shown the effect of Zinc chloride on weight loss and a decrease in blood sugar in diabetic rats. The supplementation with Zinc chloride inhibited oxidative stress and restored the liver and kidney function and structure in Streptozotocin-induced diabetes in rats. The manuscript may be accepted for publication after a few revisions.

1. Improve the quality of Figure(s) 2-6.

2. The authors must clarify why Streptozotocin was used to induce Diabetes in Rats.

3. I would like the authors to cite some information on the metal ion Copper (Cu) (apart from Zinc and Iron) as Cu is the transition metal ion closely associated with DNA base pairs, particularly guanine. 

4. Mention the bioavailability of Zn chloride in humans. Beyond what level if given extraneously it could be toxic.

5. Thoroughly check the language of the manuscript. Primarily the abstract and conclusion need to be improved.

Author Response

Reviewer 1

The authors have shown the effect of Zinc chloride on weight loss and a decrease in blood sugar in diabetic rats. The supplementation with Zinc chloride inhibited oxidative stress and restored the liver and kidney function and structure in Streptozotocin-induced diabetes in rats. The manuscript may be accepted for publication after a few revisions.

Distinguished Reviewer,

Thanking you for the time on reviewing our manuscript and the pertinent suggestions that helped us improve the content of the article, we have addressed point-by-point all the comments. Please find below the changes.

  1. Improve the quality of Figure(s) 2-6.

The quality of all figures has been improved.

 Figure 1

 Figure 2

We`ve added arrows on the images in order to highlight the histological changes encountered in the examined tissues, and we mentioned in the text.

Figure 3.

In rats treated on STZ we noticed an important alteration of the exocrine, as well as endocrine components. The beta pancreatic cells were very rare, the acinar cells appeared expanded in volume (arrows), and contained small vacuoles (v). The interlobular channels were coated with flat epithelium (Figure 3). The histopathological exam of the pancreatic fragments collected from the STZ+Zn group revealed that the atrophy of acinar cells was slighter than in STZ group, a clearer line between exocrine and endocrine areas of the pancreatic tissue being detected. A tendency to regenerate the insular areas was observed, and the vacuoles at the base of the acinar cells were few and small (Figure 3)

Figure 4

The microscopic examination of the samples obtained from hepatic tissue of STZ rats indicates liver steatosis objectified by the presence of numerous small lipid vacuoles (v) inside the hepatocytes. The hepatic cells showed moderate disturbances in their structure, their cytoplasm being slightly darker and the voluminous nuclei have prominent nucleoli. The presence of a minor periportal fibrosis and sinusoidal distension were also observed (arrows) (Figure 4).

Figure 5.

STZ administration caused important changes in kidney architecture, characterized by a medium degeneration of cortical area, tumefied nephrocytes in the tubular epithelium (T), with dark cytoplasm and hyperchromatic nuclei. Intense glomerular congestion and infiltration of mononuclear cells was also present (arrow) (Figure 5).

Figure 6.

In STZ rats, a moderate to severe myocardial congestion, consisting of moderate edema, polymorphonuclear infiltration and blood overcharge of interstitial capillaries was detected (arrows). However, there were no visible morphological changes of myocardiocytes structure (Figure 6)

  1. The authors must clarify why Streptozotocin was used to induce Diabetes in Rats.

STZ is one of the diabetogenic agents most used to induce this metabolic disturbance in rodents, because it causes the alteration of pancreatic beta cells, with a reduction in insulin levels and a decrease in glucose tolerance. The mechanisms by which STZ produces diabetes involve the deterioration of glucose oxidation and the disruption of the function of beta cells, which no longer respond to glucose. STZ shows a high affinity for lipids, being absorbed very quickly through the plasmatic membrane, and accumulated in the beta Langerhans pancreatic cells, as well as in the liver and kidney tissue, with cytotoxic effects. At the cellular level, STZ produces the glycosylation of proteins and the DNA alteration, which consequently reduces the cellular energy stores. The intensity of these effects correlates positively with the dose, the small repeated doses, producing limited structural, functional and metabolic disturbances (Yan, L.-J. the nicotinamide/streptozotocin rodent model of type 2 diabetes: renal pathophysiology and redox imbalance features. Biomolecules 2022; 12: 1225. doi: 10.3390/biom12091225). We chose to use low doses of STZ and cholesterol diet to obtain a DM model with metabolic disorders similar to human type 2 DM. In addition, STZ is easy to purchase, is not expensive, and DM type 2 appears after a relatively short time.

  1. I would like the authors to cite some information on the metal ion Copper (Cu) (apart from Zinc and Iron) as Cu is the transition metal ion closely associated with DNA base pairs, particularly guanine.

Thank you for this valuable proposal. We`ve inserted in the Introduction section some information on the metal ion copper with the corresponding references.

Copper is essential to life, being the third most abundant trace element in the body after zinc and iron. However, in adults, an amount of 80,000 to 100,000 µg of copper intakes from the outsource becomes toxic. Excess of Cu leads to damage the antioxidant enzyme function, oxidative modification of DNA and proteins, lipid oxidation, activates the redox-sensitive genes, suppresses the Zn consumption in the body, and also induces anemia by interfering with iron transport. Cu is present in every tissue of the body, being stored basically in the liver, but in smaller amounts is found in the brain, heart, kidney, and muscles. It can act as both an anti-oxidant and a pro-oxidant. Copper plays an important role in human metabolism, largely because it allows many fundamental and essential enzymes to function properly; being, also involved in the production of hemoglobin, myelin, and melanin. Thus, Cu serves as a cofactor for enzymes, including ferroxidase enzyme, cytochrome oxidase, superoxide dismutase, and amine oxidases. In gene regulation, it acts as a cofactor in a DNA binding “copper fist” protein. Due to the involvement of Cu in multiple molecular processes, especially in the course of catalytic reactions, the disruption of the homeostasis of this element is responsible for the appearance of various disorders that cause disease. Although the role of Cu is not completely deciphered, it is accepted that the labile fractions of this ion manifest their toxic effects on the cell, following the interaction with DNA and the intensification of oxidative stress, as a result of the generation of free oxygen species [2].

  1. Mustafa, S; AlSharif, M. Copper (Cu) an Essential Redox-Active Transition Metal in Living System—A Review Article. AJAC 2018; 9: 15-26. doi: 4236/ajac.2018.91002
  2. Komarnicka, U.K.; Lesiów, M.K.; Witwicki, M.; Bie´nko, A. The Bright and Dark Sides of Reactive Oxygen Species Generated by Copper–Peptide Complexes. Separations 2022; 9: 73. https://doi.org/10.3390/ separations9030073.

  1. Mention the bioavailability of Zn chloride in humans. Beyond what level if given extraneously it could be toxic.

Thank you for this valuable suggestion. We`ve included in the Introduction section some information on the bioavailability of zinc in humans, with the corresponding references.

The recommended dietary allowance (RDA) for Zn is 11 mg/day for men and 8 mg/day for women [1]. This is significantly below the LD50 value, which has been estimated to be 27 g/day in humans, based on comparison with equivalent studies in rats and mice [2]. In general, uptake of such an amount is unlikely, because approximately 225–400 mg Zn have been determined to be an emetic dose. Immediate symptoms after uptake of toxic amounts of Zn include abdominal pain, nausea, and vomiting. Additional effects include lethargy, anemia, and dizziness [3].

Clinical studies have revealed that in people with Cu deficiency, corresponding clinically objective, associated with anemia, neutropenia, and the alteration of the immune defense capacity, the administration of small doses of Zn (100-300 mg Zn/day), but also of quantities much higher than the RDA (15 mg Zn/day), important disturbances of the LDL/HDL cholesterol ratio occur. Even lower levels of Zn supplementation, closer in amount to the RDA, have been suggested to interfere with the utilization of Cu and iron and to adversely affect HDL cholesterol concentrations [4].

High-dose Zn supplementation interferes with the uptake of Cu. Hence, many of its toxic effects are in fact due to Cu deficiency. While systemic homeostasis and efficient regulatory mechanisms on the cellular level generally prevent the uptake of cytotoxic doses of exogenous Zn, endogenous Zn plays a significant role in cytotoxic events in single cells [3].

  1. Trumbo, P; Yates, AA; Schlicker, S; Poos, M. Dietary reference intakes: vitamin A, vitamin K, arsenic, boron, chromium, copper, iodine, iron, manganese, molybdenum, nickel, silicon, vanadium, and zinc. J. Am. Diet. Assoc 2001; 101, 294–301. doi: 10.1016/S0002-8223(01)00078-5.
  2. Toxicological Profile for Zinc; Agency for Toxic Substances and Disease Registry Division of Toxicology and Environmental Medicine: Atlanta, GA, USA, 2005.
  3. Plum, LM; Rink, L; Haase H. The essential toxin: impact of zinc on human health. Int J Environ Res Public Health. 2010; 7(4): 1342-1365. doi: 10.3390/ijerph7041342
  4. Hussain S, Khan M, Sheikh TMM, et al. Zinc Essentiality, Toxicity, and Its Bacterial Bioremediation: A Comprehensive Insight. Front Microbiol. 2022; 13: 900740. Published 2022 May 31. doi:10.3389/fmicb.2022.900740

  1. Thoroughly check the language of the manuscript. Primarily the abstract and conclusion need to be improved.

We’ve checked this manuscript carefully for several rounds to avoid any mistake. The manuscript has been thoroughly spell-checked and mistakes have been corrected.

The Abstract and the Conclusion sections have been improved.

Abstract. Background and Objectives: Diabetes mellitus (DM) is a complex disease, affecting the whole metabolic balance of the body and resulting in multiple organ complications: cardiovascular, neuronal, renal, etc. Our study focuses on investigating the effect of zinc chloride (Zn) on certain blood parameters suggestive for assessing the metabolic disturbances, the liver and kidney function, the oxidative stress and the immune defense capacity in experimental-induced DM with streptozotocin (STZ) and cholesterol in rats. Materials and Methods: The animals were assigned in three groups, as follows: Group 1 (Control): buffer citrate solution 0.1mL/100g body; Group 2 (STZ): 20 mg/kg body STZ and fat diet (10 g cholesterol/100 g diet); Group 3 (STZ+Zn): 20 mg/kg body STZ + 5 mg/kg body Zn chloride and the same fat diet. DM was induced by administering STZ in a single take daily, for three consecutive days, Zn and citrate buffer were administered orally for a month. The protocol was approved by the Ethics Committee of University ‘Grigore T Popa’ Iasi, in agreement with the International Regulations about the handling of laboratory animals. Results: The use of STZ in rats fed with cholesterol was correlated with an important weight gain, hyperglycemia, the intensification of the transaminases activity, increase in serum alkaline phosphatase, cholesterol, triglyceride, urea, creatinine and in malondialdehyde. Conclusions: The treatment with Zn resulted in weight loss and a decrease in blood sugar of diabetic rats. Supplementation with Zn notably reduced oxidative stress, preserved the pancreatic architecture, and restored the liver and kidney function and structure in STZ-induced DM in rats.

Conclusions: The results obtained by us are of great importance in medical practice, regarding the favorable effects of Zn in slowing down the evolution of the disease and the occurrence of organic complications in diabetes.

We are grateful for the time and effort spent in providing constructive feedback as to how we can improve our paper. The text of the manuscript has been enriched and substantially enhanced, the alterations are visibly highlighted and we now believe that it fulfils criteria for publication in your well-respected journal.

Yours sincerely,

Liliana Mititelu-Tartau, corresponding author

Reviewer 2 Report

In the manuscript entitled Zinc Chloride Enhances the antioxidant status, improving the functional and structural organic disturbances in Streptozotomicin-Induced diabetes in rats”, authors develop some experimental study in order to clarify or confirm the effect of zinc chloride on the biochemical changes and oxidative stress in diabetic rats. The topic would be of interest and worth to be furthered, however, manuscript presents some critical issues described below:

-the section 2.1 is too hasty, only three substances were cited

-section 2.3.2 is not fully exhaustive, haematological and biochemical parameters analysis are not clearly described

-figure 4, 5, 6 have to be described more in details also adding arrows on image in order to underline the histologic modification exerted by STZ and/or Zn treatment.

-most of the discussion is focused on previous data regarding the protective effect exerted by Zinc treatment on the onset of diabetic complications, while only few words is used by Authors to comment their results. In addition, STZ used in these experimental conditions not provide any variation in the leukocite formula components, bilirubin concentration or in the serum complement activity and in the NBT test: the STZ used in these experimental conditions could not be enough explanatory on the complete onset of diabetes and its complications. Could Zinc protect the organism only in spite of a mild manifestation of the disease? The doubt remains and Authors should add some considerations about it.

Author Response

Reviewer 2

In the manuscript entitled “Zinc Chloride Enhances the antioxidant status, improving the functional and structural organic disturbances in Streptozotocin-Induced diabetes in rats”, authors develop some experimental study in order to clarify or confirm the effect of zinc chloride on the biochemical changes and oxidative stress in diabetic rats. The topic would be of interest and worth to be furthered, however, manuscript presents some critical issues described below:

Distinguished Reviewer,

We are very grateful for your constructive comments and also for the time and effort on reviewing our manuscript. We have addressed point-by-point all the comments and made corrections in our manuscript using tracked changes.

- the section 2.1 is too hasty, only three substances were cited

Thank you very much for your comment. We mentioned all the substances used in the experiment, with their characteristics and catalogue name.

Zn chloride (catalogue name 208086, molecular weight MW: 136.3), streptozotocin (STZ) (catalogue name S0130, ≥75% α-anomer basis, ≥98% HPLC, MW: 265.22), cholesterol (catalogue name C8667, ≥99%, MW: 386.65), buffer citrate (catalogue name C2488, citrate buffer solution 0.09 M) and distilled water (catalogue name 07-6061, MW: 18.02) were purchased from Sigma-Aldrich, Chemical Co (Steinheim, Germany) (www.sigma-aldrich.com).

- section 2.3.2 is not fully exhaustive, haematological and biochemical parameters analysis are not clearly described

As advised, we`ve given the details about the hematological and biochemical analysis parameters.

The number of erythrocytes was estimated with the help of the automatic hematology analyzer during their passage through an orifice through which they are directed in a single row by the hydrodynamic focusing method. Differential white blood cells counts were evaluated in all rats through the direct current detection technique, and percentages were assessed using a the specific analyzer (Fischbach F. Blood Studies: hematology and coagulation; Appendix J: Effects of the most commonly used drugs on frequently ordered laboratory tests. In A manual of laboratory and diagnostic tests. Lippincott Williams & Wilkins, Philadelphia, 8 ed. 2009, 67-110, 1227-1247. ISBN: 978-1-4511-9089-2).

The transaminases activity was measured on blood collected in vacutainers without anticoagulant by the IFCC (International Federation for Clinical Chemistry) standardized kinetic method with pyridoxal phosphate.

The serum levels of biochemical parameters were measured on blood samples collected without anticoagulant, using spectrophotometric methods (colorimetric for liver transaminases, total bilirubin, alkaline phosphatase and creatinine; colorimetric/enzymatic for triglycerides, total cholesterol, and its fractions; respectively kinetic for urea) (Fischbach F. Chemistry studies. In A manual of laboratory and diagnostic tests. Lippincott Williams & Wilkins, USA, 8 ed., 2009, 452-455. ISBN: 978-1-4511-9089-2).

- figure 4, 5, 6 have to be described more in details also adding arrows on image in order to underline the histologic modification exerted by STZ and/or Zn treatment.

                Thank you very much for your suggestion. We`ve modified the mentioned figures, adding arrows on the images in order to highlight the histological changes encountered in the examined tissues.

Figure 3.

In rats treated on STZ we noticed an important alteration of the exocrine, as well as endocrine components. The beta pancreatic cells were very rare, the acinar cells appeared expanded in volume (arrows), and contained small vacuoles (v). The interlobular channels were coated with flat epithelium (Figure 3). The histopathological exam of the pancreatic fragments collected from the STZ+Zn group revealed that the atrophy of acinar cells was slighter than in STZ group, a clearer line between exocrine and endocrine areas of the pancreatic tissue being detected. A tendency to regenerate the insular areas was observed, and the vacuoles at the base of the acinar cells were few and small (Figure 3)

Figure 4

The microscopic examination of the samples obtained from hepatic tissue of STZ rats indicates liver steatosis objectified by the presence of numerous small lipid vacuoles (v) inside the hepatocytes. The hepatic cells showed moderate disturbances in their structure, their cytoplasm being slightly darker and the voluminous nuclei have prominent nucleoli. The presence of a minor periportal fibrosis and sinusoidal distension were also observed (arrows) (Figure 4).

Figure 5.

STZ administration caused important changes in kidney architecture, characterized by a medium degeneration of cortical area, tumefied nephrocytes in the tubular epithelium (T), with dark cytoplasm and hyperchromatic nuclei. Intense glomerular congestion and infiltration of mononuclear cells was also present (arrow) (Figure 5).

In STZ rats, a moderate to severe myocardial congestion, consisting of moderate edema, polymorphonuclear infiltration and blood overcharge of interstitial capillaries was detected (arrows). However, there were no visible morphological changes of myocardiocytes structure (Figure 6)

- most of the discussion is focused on previous data regarding the protective effect exerted by Zinc treatment on the onset of diabetic complications, while only few words is used by Authors to comment their results. In addition, STZ used in these experimental conditions not provide any variation in the leukocyte formula components, bilirubin concentration or in the serum complement activity and in the NBT test.

Thank you very much for your constructive comment. The Discussion section has been significantly improved, according to your suggestions, by adding more comments about our results.

The present research revealed no substantial changes in the number of erythrocytes or in the percentage of neutrophils, lymphocytes, eosinophils, monocytes and basophils in diabetic rats and STZ+Zn group when compared to control animals. Moreover, the percentage of these leukocyte formula components was not significant modified at the two time moments of evaluation, compared to baseline. Literature data reported considerable variations in the percentages of these elements in type 1 diabetes, induced by a single high dose of STZ in rats (Yakhchalian H, Mohammadian N, Hatami K, Nosrati H, Yousofvand N. Hematological and serum biochemical analysis of streptozotocin-induced insulin dependent diabetes mellitus in male adult wistar rats. bioRxiv; 2018; 359844 1-11. doi: https://doi.org/10.1101/359844)

Concerning the effects on hematological parameters of low doses of STZ, the communicated results were only few and controversial. Some authors demonstrated a significant decrease in both neutrophils and lymphocytes percentages, as well as in the number of RBC, in rats subjected to 15 mg/kbw STZ compared normal rats (Hajam YA, Rai S, Ghosh H, Basheer M. Combined administration of exogenous melatonin and insulin ameliorates streptozotocin induced toxic alteration on hematological parameters in diabetic male Wistar rats. Toxicol Rep. 2020; 7: 353-359. doi: 10.1016/j.toxrep.2020.01.020). Other researchers proved a marked increase in the neutrophils percentage, and a diminution in lymphocytes percentage and in the RBC number in type 2 diabetes experimentally induced with 35 mg/kg STZ and fat diet in rats when compared with animals treated with buffer citrate (Mahmoud AM. Hematological alterations in diabetic rats - Role of adipocytokines and effect of citrus flavonoids. EXCLI J. 2013;12:647-657. PMID: 26966427; PMCID: PMC4778348.)

                Laboratory analysis did not evidence the modifications in the serum levels of LDL-cholesterol, even a slight increase was noted after 4 weeks in diabetic rats, compared to control group. Total bilirubin values in blood were not extensive influenced by the low doses of STZ, which probably induced only moderate alterations in liver functions.

                The STZ-induced type 2 DM in rats with high fat diet was not accompanied by essential changes in the activity of serum complement and in the phagocytic capacity of peripheral neutrophils, which denotes the fact that, with the dose used, this diabetogenic agent does not profoundly alter the immune defense capacity.

- the STZ used in these experimental conditions could not be enough explanatory on the complete onset of diabetes and its complications. Could Zinc protect the organism only in spite of a mild manifestation of the disease? The doubt remains and Authors should add some considerations about it.

The research carried out clearly shows the fact that the treatment with cholesterol and 15 mg/kg STZ, 3 consecutive days, caused DM type 2 in rats, characterized by weight gain and hyperglycemia, and was correlated with moderate structural and functional liver and renal disturbances, as well as an intensification of oxidative stress, changes that could be present in mild and moderate phases of the disease.

The fact that Zn supplementation contributes to reducing weight, normalizing blood glucose levels, reducing oxidative stress and ameliorating liver and kidney damage constitutes an argument regarding the protective role of this trace element in mild and moderate forms of DM. Regarding the protective influence of Zn in severe forms of DM, further in-depth studies are required.

We are grateful for the time and effort spent in providing constructive feedback as to how we can improve our paper. The text of the manuscript has been enriched and substantially enhanced, the alterations are visibly highlighted and we now believe that it fulfils criteria for publication in your well-respected journal.

Yours sincerely,

Liliana Mititelu-Tartau, corresponding author

Round 2

Reviewer 1 Report

The manuscript can be accepted for publication.

Reviewer 2 Report

After Authors revision, the manuscript can be accepted.